# Miniaturized Multi-Port Microstrip Patch Antenna Using Metamaterial for Passive UHF RFID-Tag Sensor Applications

**DOI:** 10.3390/s19091982

**Published:** 2019-04-28

**Authors:** Jamal Zaid, Abdulhadi E. Abdulhadi, Tayeb A. Denidni

**Affiliations:** Institut national de la Recherche scientifique Centre, Centre Énergie Matériaux Télécommunications 800, De La Gauchetière Ouest Bureau 6900, Montréal, QC H5A 1K6, Canada; abdulhadi.abdulhadi@mail.mcgill.ca (A.E.A.); denidni@emt.inrs.ca (T.A.D.)

**Keywords:** self-powered, multi-port patch antenna, RFID tag based-sensor, miniaturization, magneto-dielectric substrate, reading range, minimum required power

## Abstract

In this paper, a miniaturized Ultra High Frequency Radio Frequency Identification (UHF-RFID) tag-based sensor antenna using a magneto- dielectric substrate (MDS) for wireless identification and sensor applications is presented. Two models of RFID tag-based sensors are designed, fabricated and measured. The first model uses two RFID tags; both of the tags are incorporated with two RFID chips. A passive sensor is also integrated in one of the proposed tags to serve as a sensor node, while the other tag is used as a reference node. Based on the difference in the minimum power required to activate the reference and sensor nodes, the sensed data (temperature or humidity) can be determined. The magneto-dielectric substrate layer is placed underneath the patch antenna to reduce the size of the proposed sensor by about 75% compared to a conventional RFID tag-based sensor. The magneto-dielectric layer is thin enough to embed in the planer circuit. To reduce the size of the proposed sensor, a multi-port tag for including the reference and sensor node in one antenna is also presented. The proposed RFID tag-based sensors have several features such as small size, they are completely capable for two objectives at the same time and easy to integrate with a planer circuit.

## 1. Introduction

In the recent decade, wireless communication network applications have increased, particularly for Radio Frequency Identification (RFID) applications, such as security systems, asset tracking, remote wireless identification, access control and wireless sensor networks (WSNs). Consequently, RFID tag-based sensors have become an extremely attractive research topic among researchers due to their use for identification and sensing at the same time, low consumption power, long lifetime and low cost as discussed in [1,2,3,4]. Various RFID-tag based sensors have been proposed in the literature as described in [1,2,3,4,5,6,7,8,9,10,11,12]. The operation mechanism of the RFID tag-based sensor uses two nodes, one node is employed as a reference node and the second node is used as a sensor node. These nodes generate the required operating power from the same signals received from the RFID reader. In the sensor node circuit, a passive sensor is used to measure physical quantities such as temperature, humidity, pressure, etc. The other node is connected with a RFID chip and used only as a reference. Since both nodes are activated by the received signal from the reader, any change in the temperature or humidity will introduce a mismatch between the antenna and the sensor node. Thus, the sensor node will require more power to activate the RFID chip compared to the reference node. Then, the sensed physical quantity can be measured using a commercial RFID reader from the ratio of the measured power of the sensor and reference nodes. In previous sensor systems, the tag antennas are often an omnidirectional antenna, such as a dipole, folded loop and meander monopole antenna [1,2,3,4,5,6,7,8], [10] and [11], respectively. Although, these systems are self-powered, the omnidirectional antennas cannot be mounted on a metallic surface because the performance of the antenna will be degraded, including the radiation pattern, resonance frequency, and efficiency. Therefore, the sensor will not be able to give accurate results. Other systems have used a half omnidirectional antenna such as a patch antenna to solve the previous problem. Thus, the patch antenna can be mounted on a metallic surface without any performance degradation, as mentioned in our previous works [12,13,14]. Some systems have used a multi-port antenna, including the reference node and the sensor node in the same antenna, instead of using two antennas as reported in [12,13,14]. However, this approach has some disadvantages, such as the large size of the antenna and using a battery, microcontroller and solar cells [12,13].

To overcome the previous problems, we propose a new tag antenna that operates at the North America frequency band: 902–928 MHz. The proposed tag antenna is matched with two RFID chips, one of them is represented as a reference node, and the another one is soldered parallel with a resistive sensor to represent a sensor node. The proposed tag antenna can be placed on the metallic surface, a magneto-dielectric substrate is used to achieve miniaturization of a microstrip patch antenna of about 75% the size of the conventional patch antenna. However, the antenna fabrication will need more time and accuracy, and the proposed antenna can be used instead of two antennas. Moreover, this antenna is the smallest compared with previous works [12,13,14]. Two cases are studied in this paper. The first case uses two antennas: one for the reference node and the second one for the sensor node. The second case uses a multiport antenna including the reference node and sensor node in the same antenna. A variable resistor (10, 20, 50, 100, 200 Ω) is used rather than the resistive sensor in both cases. Both antennas were fabricated, measured, and the obtained results are presented and compared. 

The paper is organized as follow: Section 2 is a review of RFID tag-based sensors. The proposed unit cell is presented in Section 2. Section 3 explains the proposed antenna design. The single port tag is designed, fabricated and measured in Section 4. The single port tag-based sensor is designed, fabricated and measured in Section 5. In Section 6, a multi-port tag-based sensor is designed and measured. The conclusion is reported in Section 7.

## 2. Proposed Unit Cell

Since microstrip patch antennas have many advantages such as they are relatively inexpensive, easy to fabricate and able to be integrated with planar circuits, they have become more popular in wireless communication networks. However, microstrip patch antennas have a narrow bandwidth and relatively large size. The size of a microstrip patch antenna is inversely proportional to the product of dielectric permittivity and magneto permeability. Based on Equations (1) and (2), the product of effective permeability and permittivity is inversely proportional to the resonance frequency. Therefore, by increasing the product of the effective permeability and permittivity, the resonance frequency can be easily shifted to lower frequencies [15].
(1)$$f=\frac{c}{2\left(l+h\right)\sqrt{\mu_{e}\varepsilon_{e}}},$$
(2)$$\varepsilon_{e}=\frac{\varepsilon_{r+1}}{2}+\frac{\varepsilon_{r-1}}{2}{\left(1+\frac{12h}{w}\right)}^{-0.5},$$

The split ring resonator (SRR) has been used for miniaturizing the antenna size as proposed in [16,17]. Based on this, the magnetic field perpendicular to the ring (loop) induces a current in the ring, as a result, the induced current will create magneto dipoles. Consequently, the product of permeability and permittivity in the substrate will be increased. This technique can compensate for degradation of the bandwidth, which leads to miniaturizing the antenna size. However, in previous works [16,17], the metamaterial substrate layers were thick and cannot be used in a planar circuit whereas the proposed magneto-dielectric substrate is thin enough to use in the planar circuit. As shown in Figure 1, under specific boundary conditions, zy-plane walls are perfect magnetic conductors (PMCs), xy-plane walls are perfect electric conductors (PECs) and zx-ports, and the effective parameters of the proposed unit cell are extracted using the method reported by Shi et al. as shown in Figure 2 [18]. The magneto-dielectric substrate layer is composed of 34 unit cells (2 × 17) and it is placed underneath the patch. The unit cell is printed on the substrate (RO4003 with *ε_r_* = 3.55, and tan *δ* = 0.0027), and is composed of three layers of the substrate with different thickness: lower substrate (*hl* = 0.813 mm), middle substrate (*hm* = 0.203) and upper substrate (*hu* = 0.203 mm). The lower substrate consists of two vias, ground and strip (PEC) on the top. The middle substrate consists of one via and strip on the top. The upper substrate consists of (PEC) on the top, which presents a patch antenna. The product of the magnetic permeability can be controlled by changing the intersection area between the strips, which works as a capacitor or changing the length of vias, which works as an inductor. In the trade-off between high product permeability and permittivity and the losses, the best value for the product of the magnetic permeability and electric permittivity is around 16 at 915 MHz and the loss is roughly zero at the same frequency.

## 3. Antenna Design 

Using the proposed magneto-dielectric substrate, the patch antenna is miniaturized about 75% compared to a conventional patch printed on the electric substrate (RO4003 with *ε_r* = 3.55, and tan*δ* = 0.0027). The size of the conventional patch printed on the electric substrate is (*L* = 108.68, W = 86.81 and h = 1.219 all dimensions are mm) and the proposed antenna size is (*L* = 47, W = 48 and h = 1.219 all dimensions are mm). Consequently, by comparing the proposed antenna with the conventional antenna in terms of size, the proposed antenna is reduced in size by 75%.

Since the RFID tag generates the activated power using the received signal from the reader, the tag-antenna and the matching between the tag-antenna and the RFID chip play an important role in the performance of the whole system. The patch antenna is matched with an RFID chip by inductive loop matching to make a conjugate matching (Z*chip = Z Antenna). Thus, most of the received power from the reader will be transferred to the RFID chip and vice versa. The input impedance of the RFID chip (RI-UHF-IC116-00) at 915 MHz is equal to 25 − j190 and the conjugate impedance is 25 + j190. The RFID chip is presented by the lumped port (24 Ω) and capacitor (0.95 pF) as equivalent elements in HFSS. 

Figure 3 shows the return loss and radiation pattern of the proposed antenna at the desired frequency (915 MHz) is about −16 dB, which proves very good matching between the antenna and the loop. Figure 5 shows the fabricated antenna A, which is matched with one RFID chip. All antenna dimensions (*L* = length of the patch, *W* = width of the patch, *Lw* = width of the loop, *Ll* = length of the loop, d = distance between the loop and the patch and *t* = width of the loop line), are *L* = 48, *W* = 47, *Lw* = 4.8, *Ll* = 34, *d* = 0.2, and *t* = 0.4, (all dimensions are mm).

## 4. Single Port Antenna 

The patch antenna is placed on the magneto-dielectric substrate that is composed of 34 unit cells (2 × 17), as illustrated in Figure 4. The single port antenna (antenna A) in Figure 5 is designed and fabricated using a magneto-dielectric substrate. The inductive loop technique is employed to match the patch antenna and the RFID chip. By changing the dimensions of the inductive loop (length and width), the imaginary part can be controlled and by changing the distance between the patch antenna and the inductive loop, the real part can be adjusted to obtain the matching. The proposed antenna and the RFID reader (GAO 216010) are adjusted in the line of sight, then the distance between them is gradually increased until the maximum reading range is obtained, as shown in Figure 6. The maximum reading range is about 11 m at 937 MHz due to permittivity tolerance of ± 0.05 and fabrication accuracy; the resonance frequency is shifted about 22 MHz from 915 MHz to 937 MHz as illustrated in Figure 7. Although the RFID chip has activated power about −20 dBm, the activated power generated from the received power by the tag depends on the antenna performance and the matching between the antenna and RFID chip. The minimum required power for activating the RFID chip is measured and plotted in Figure 8. From this figure, the minimum power required from reader is roughly 13 dBm at 937 MHz.

## 5. Single Port Antenna-Based Sensor 

Some systems in the literature have used two dipole antennas, one as a sensor node and another as a reference node, as reported in [1,2,3,4,5,6,7,8]. Because the dipole is omnidirectionally radiated, it cannot be mounted on the metallic surface. Moreover, it is influenced by the human body as discussed in our previous work [14]. The patch antenna has been used in RFID-tag based sensors as reported in [12,13,14]. However, the size of the patch antenna is larger. For this reason, in this work, a magneto-dielectric substrate layer is placed underneath the patch for miniaturizing the proposed antenna and obtaining the smallest antenna size compared to previous works [12,13,14]. To use the RFID-tag as a sensor, the resistive sensor (variable resistance) is connected parallel with the RFID chip as shown in Figure 9 (antenna B). Since the climate room unavailable within measurement time, the variable resistance is used instead of the resistive sensor. Antenna A is used as a reference node and antenna B as a sensor node. By changing the variable resistor (10, 20, 50, 100 and 200 Ω) so the matching between antenna B and the RFID chip will be changed as well as the minimum required power. However, the minimum required power for activating the reference node will not be changed at all values of resistance because the matching does not change. The minimum required power versus frequency is measured at each value of the resistor (10, 20, 50, 100 and 200 Ω) and plotted in Figure 10. As a result of this, the minimum required power is inversely proportional to the resistor value as illustrated in Figure 10. The maximum reading range is measured at all resistances, as shown in Figure 11. The minimum required power is increased by decreasing the resistance value. The minimum required power to activate the sensor node is 25.36 dBm, 23.5 dBm, 21.72 dBm, 19.53 dBm and 17.56 dBm at resistor values 10 Ω, 20 Ω, 50 Ω, 100 Ω and 200 Ω, respectively. The minimum power at the reference node is constant (13 dBm) at all resistance sensor values. By comparing the minimum power of the reference node at 937 MHz with the minimum power of the sensor node with the different values of the resistor at the same frequency as shown in Figure 12, we can determine the temperature or humidity.

## 6. Multi-Port Antenna-Based Sensor 

In the previous section, two antennas were used in the RFID-tag based sensor, one as a reference node and one as a sensor node. Some previous systems have used a multi-port antenna, thus, the reference node and sensor node are included in one antenna [12,13,14]. However, a multi-port antenna-based sensor has more advantages, such as low cost, and the feature of the sensor node and reference node with the same received signal. 

The reader leads to both nodes is in the same environment, but the size of the antenna is still large. Therefore, the proposed antenna is a miniaturized multi-port antenna. The patch antenna is placed on the magneto-dielectric substrate layer, and it is matched with two RFID chips using an inductive match loop. One of the RFID chip is used as a reference node and the second one is connected parallel with the resistive sensor (variable resistor) as a sensor node, as shown in Figure 13 (antenna C). The variable resistor is changed from 10, 20, 50, 100 to 200 Ω and the minimum required power at the reference and sensor nodes against frequency is measured at each value. The minimum required power to activate the sensor node at 937 MHz is 35.79, 33.8, 32.03, 29.16 and 27.15 dBm when the resistive sensor is 10, 20, 50, 100 and 200 Ω, respectively, as illustrated in Figure 14. However, the minimum required power for the reference node is almost the same in all cases, it is about 44 dBm as can be seen in Figure 15. The minimum power for activating the sensor node against resistance at 937 MHz is plotted in Figure 16. From the difference in activated power, the temperature or humidity can be extracted. The minimum power of multi-port antenna (C) is higher than the minimum power of single port antenna (B) due to coupling between the two ports.

## 7. Conclusions

In this paper, a miniaturized UHF RFID tag-based sensor has been presented. The proposed antenna is placed on a magneto-dielectric substrate for miniaturizing antenna size. The magneto-dielectric substrate consisted of 34 unit cells that are arranged in tow arrows. Two sensing methods have been designed and fabricated, one uses two antennas and the other one antenna with a multi-port antenna. The proposed antenna was matched with a RFID chip using an inductive loop matching technique, and the resistive sensor was connected parallel with a RFID chip. Both RFID tag-based sensors have been measured with different resistor values (10, 20, 50, 100, 200 Ω). The proposed antenna covers the North America operating frequency band, and it can be used for identification and sensing wireless applications. The proposed sensors can be mounted on a metallic surface without any influence. Moreover, the size is reduced by about 75% compared with a conventional antenna. The proposed tag-sensor depends on the difference in minimum required power to activate the sensor node and the reference node. Based on the difference in the required minimum power between the nodes, the proposed sensor can be used to determine humidity or temperature value. Therefore, the proposed antenna can be used for identification and as a sensor. The simulated and measured results have shown good agreement. 

## Figures and Tables

**Figure 1 sensors-19-01982-f001:**
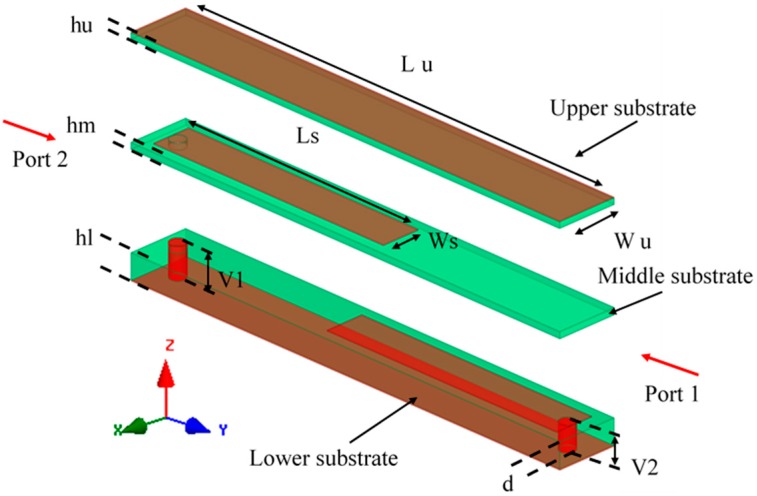
The proposed unit cell under specified boundary condition. The unit cell physical dimensions are *Wu* = 2.76, *Lu* = 23.5, *Ws* = 1.8, *Ls* = 13.65, *d* = 0.7, *hl* = 0.813, *hm* = 0.203, *hu* = 0.203. (all dimensions in mm).

**Figure 2 sensors-19-01982-f002:**
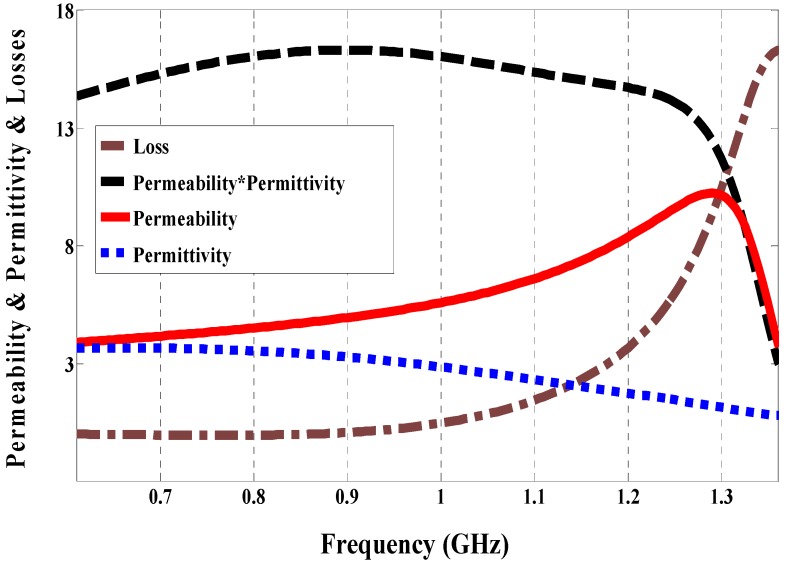
Effective constitutive parameters of the proposed unit cell. The dashed-dotted line shows the sum of the magnetic and electric losses, the dashed line shows the product of dielectric permittivity and magnetic permeability, the solid line shows magnetic permeability, and the dotted line shows electric permittivity.

**Figure 3 sensors-19-01982-f003:**
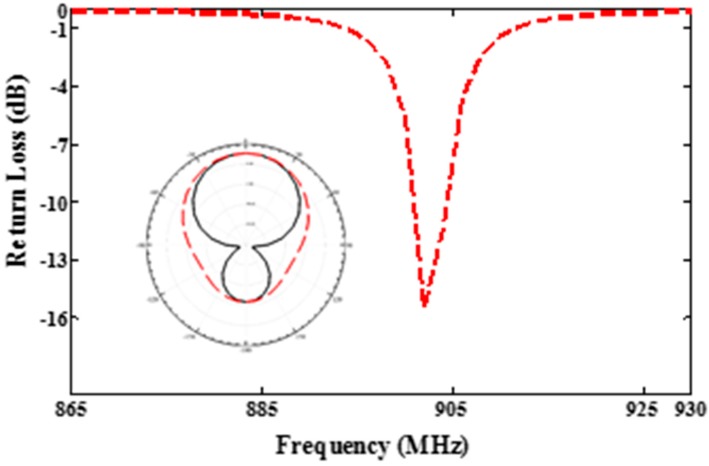
Return loss of the proposed antenna and radiation pattern.

**Figure 4 sensors-19-01982-f004:**
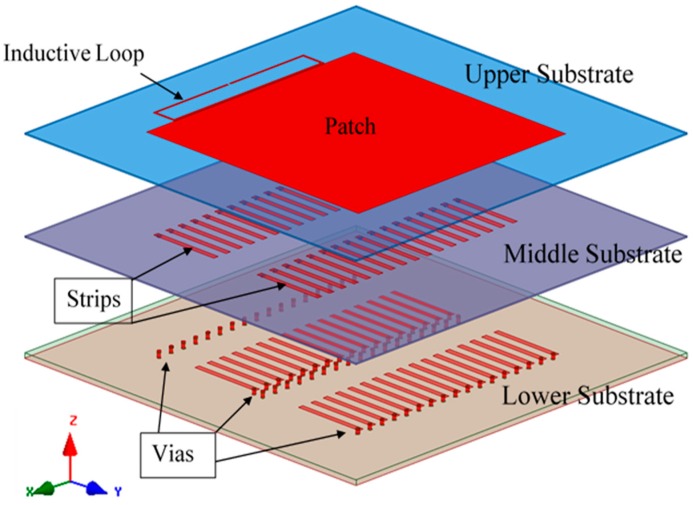
Structure of the proposed antenna.

**Figure 5 sensors-19-01982-f005:**
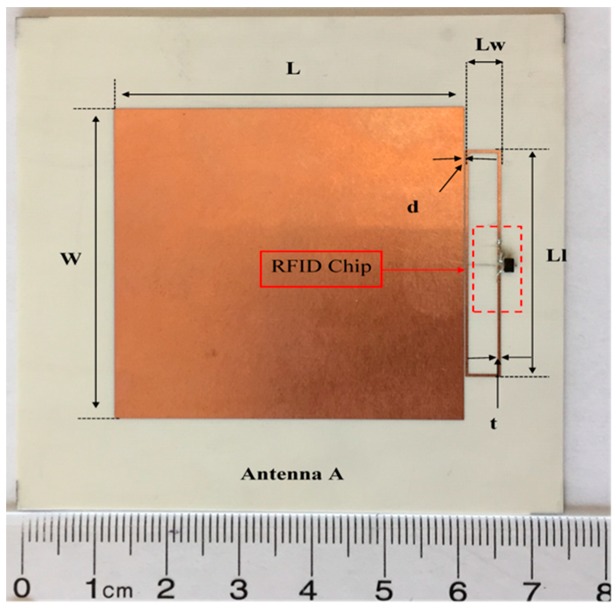
Fabricated antenna. The proposed antenna dimensions are *L* = 48, *W* = 47, *Lw* = 4.8, *Ll* = 34, *d* = 0.2, and *t* = 0.4 (all dimensions are mm).

**Figure 6 sensors-19-01982-f006:**
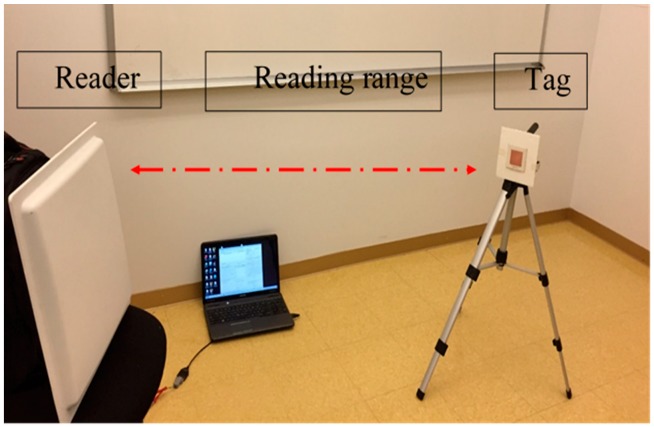
Measured reading range.

**Figure 7 sensors-19-01982-f007:**
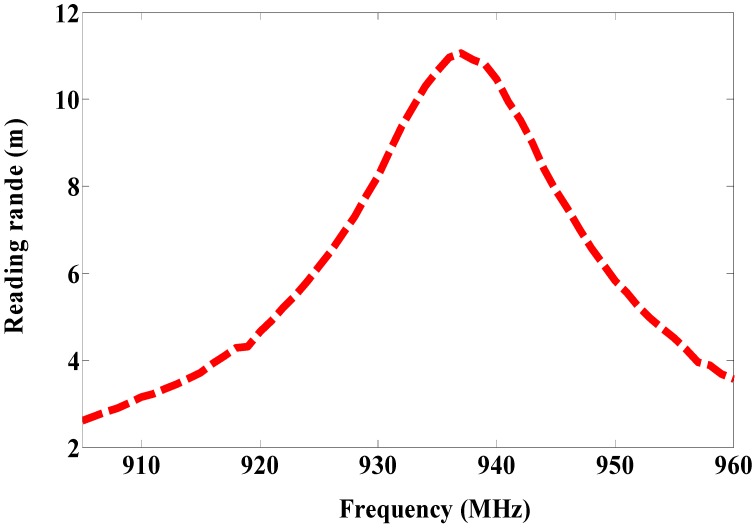
Measured reading range of antenna A.

**Figure 8 sensors-19-01982-f008:**
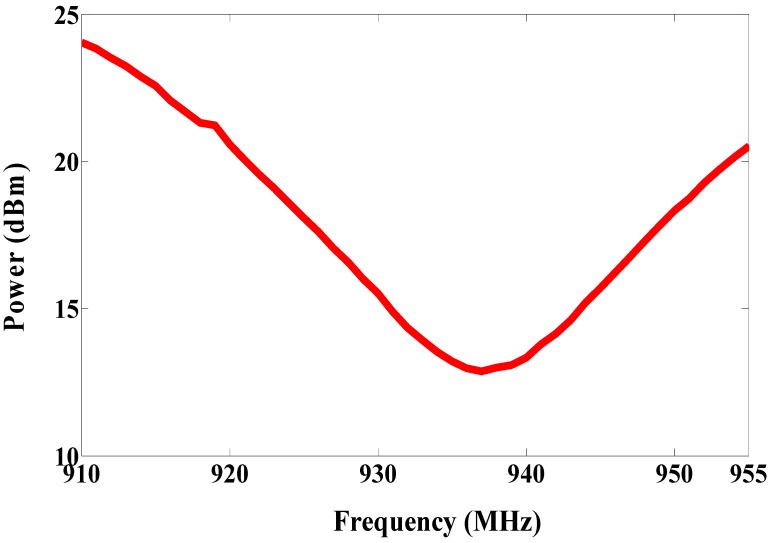
Measured minimum required power for activating antenna A RFID chip.

**Figure 9 sensors-19-01982-f009:**
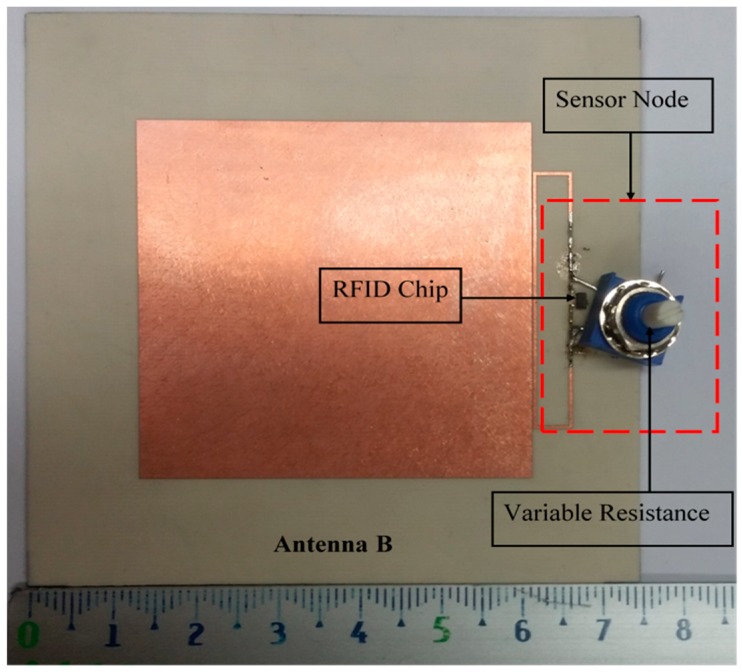
Fabricated antenna with resistive sensor (antenna B).

**Figure 10 sensors-19-01982-f010:**
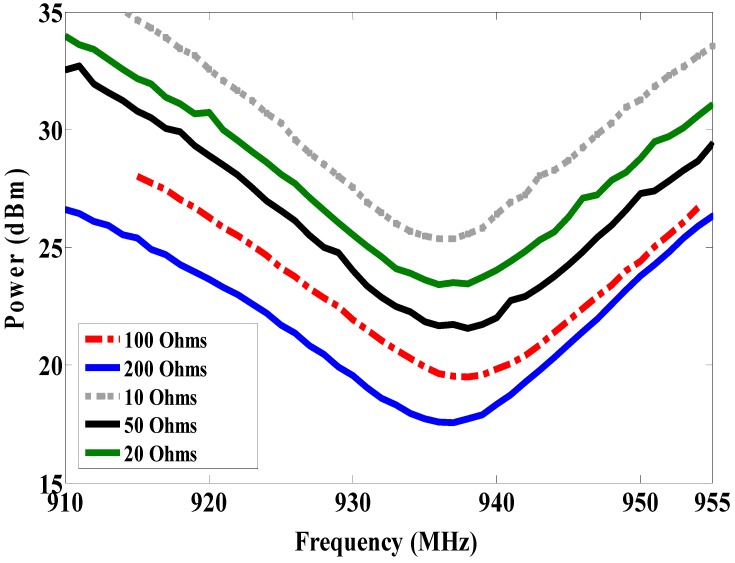
Measured minimum required power to activate sensor node with different resistance (antenna B).

**Figure 11 sensors-19-01982-f011:**
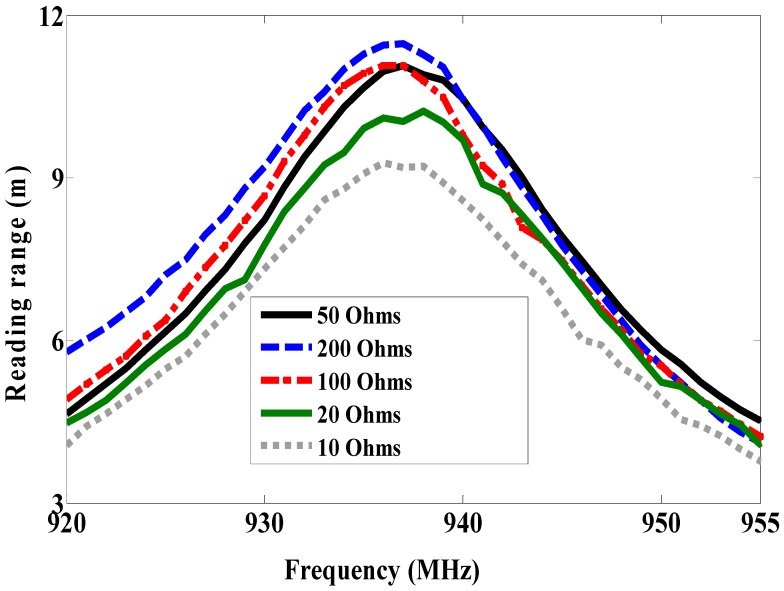
Measured reading range with different resistance (antenna B).

**Figure 12 sensors-19-01982-f012:**
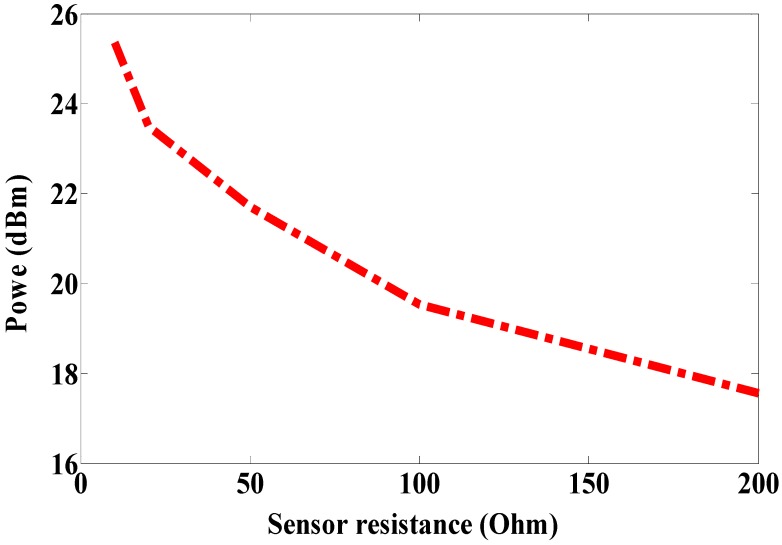
Measured minimum power to activate sensor node at 937 MHz (antenna B).

**Figure 13 sensors-19-01982-f013:**
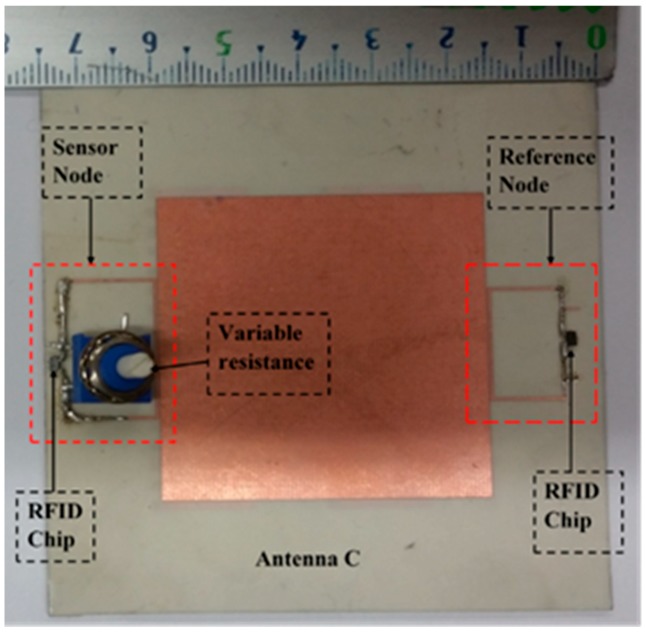
Fabricated multi-port antenna (antenna C).

**Figure 14 sensors-19-01982-f014:**
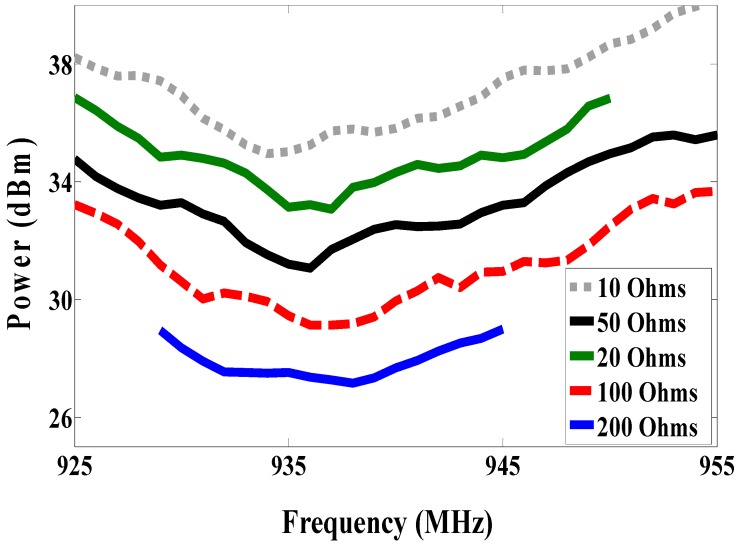
Measured minimum power required to activate sensor node with different resistance (antenna C).

**Figure 15 sensors-19-01982-f015:**
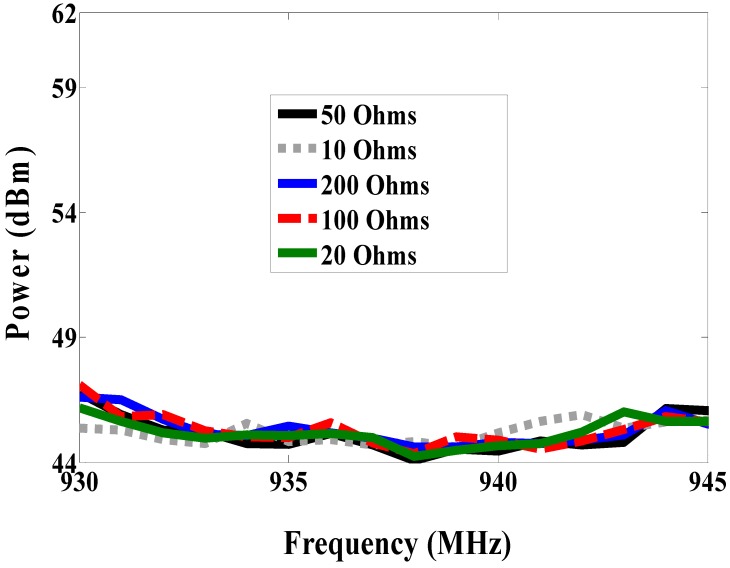
Measured minimum power required to activate reference node with different resistance (antenna C).

**Figure 16 sensors-19-01982-f016:**
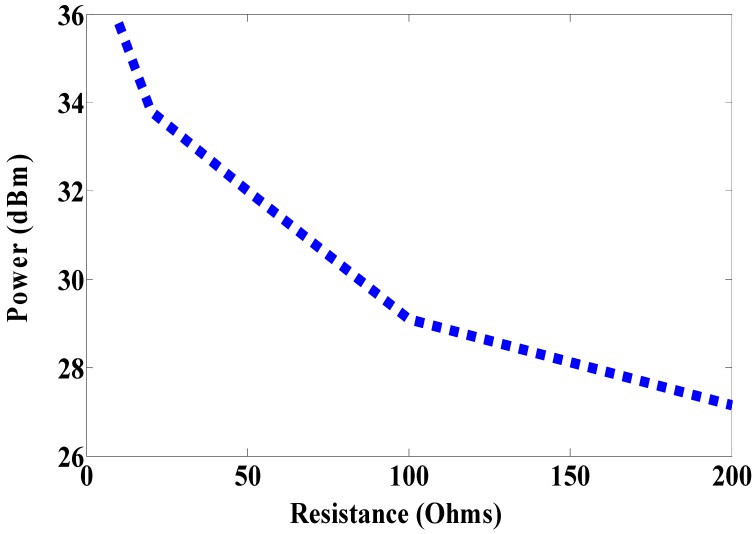
Measured minimum power to activate sensor node at 937 MHz (antenna C).

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
