# Peer review of "Miniaturized Multi-Port Microstrip Patch Antenna Using Metamaterial for Passive UHF RFID-Tag Sensor Applications"

_sensors, 2019, doi:10.3390/s19091982_

Reviewer 1 Report

The paper promisses to provide two novel contributions: i) "metamaterial" magnetodielectric substrate enabling 75% reduction of patch antenna size, ii) dual-port antenna with sensor and reference nodes in the same antenna. However significant deficiencies in technical soudness seems to be present in the text.

Comments/questions:

1.) The authors claim in the title that they use "metamaterial" and in the section 2. is entitled "Proposed unit cell" however any equivalent circuit or numerical simulation models of the unit cell for extraction of substrate effective parameters proving the claim "product of magnetic permeability and electric permitivity is around 16" is not provided. Thus claim on metamaterial character of the substrate is not supported by any analysis or measurement. Further the claim of 75% reduction of patch antenna size is not supported by the comparison of the resonant frequencies of the proposed and conventional patch antenna. This part of the paper is thus not technically sound and needs to be significantly improved.

2.) Parallel connection of variable resistor to the RFID chip changes the matching between chip and antenna input impedance resulting in changes in minimum required power for activating the chip and  changes in read distance. How this change might be used for reliable sensing if real application of RFID tag must include multipath propagation (at least ray reflected from the ground) which causes ripple on the received power level and thus affect received power and reading distance. Has such an effect been considered and quantitatively evaluated? If not any analysis have to support eventual low or negligible effect of the multipath propagation on the sensing capability.

3.) The text needs improvements: a) instead of word "Ohms" use Greek letter, b) variables are to be in italic, c) p. 4, r. 152 - "MHz" is missing behind the number 937.

4.) Further, the term "self-power"in the text seems to me unjustified and redundant independently on the fact that it has already been used by the authors in ref. [12-13]. The antenna tag does not power itself. It is powered by the energy of incident electromagnetic wave. Established RFID terminology uses for such transponders the term "passive" transponders; see e.g. ref. [1-3],[6-7]. Please justify the use of the proposed term or avoid it.

Author Response

Responses to the reviewers’ comments for the manuscript
Miniaturized Multiport Microstrip Patch Antenna Using Metamaterial for Passive UHF RFID-Tag Sensor Applications

We would like to thank the reviewers for their insightful comments regarding our manuscript. Their valuable input has helped us to improve its quality and readability. We have addressed all the reviewers’ comments in this revised version; our detailed responses to the reviewers’ comments are presented below.

REVIEWERS’ COMMENTS:

Reviewer 1:

Comments to the Author:

1.)    The authors claim in the title that they use "metamaterial" and in the section 2. is entitled "Proposed unit cell" however any equivalent circuit or numerical simulation models of the unit cell for extraction of substrate effective parameters proving the claim "product of magnetic permeability and electric permitivity is around 16" is not provided. Thus claim on metamaterial character of the substrate is not supported by any analysis or measurement. Further the claim of 75% reduction of patch antenna size is not supported by the comparison of the resonant frequencies of the proposed and conventional patch antenna. This part of the paper is thus not technically sound and needs to be significantly improved.

We would like to manifest our appreciation for the careful attention of the reviewer.

We have corrected and added more details for this part in the revised paper (please see pages 2 and 3).

2.)     Parallel connection of variable resistor to the RFID chip changes the matching between chip and antenna input impedance resulting in changes in minimum required power for activating the chip and changes in read distance. How this change might be used for reliable sensing if real application of RFID tag must include multipath propagation (at least ray reflected from the ground) which causes ripple on the received power level and thus affect received power and reading distance. Has such an effect been considered and quantitatively evaluated? If not any analysis have to support eventual low or negligible effect of the multipath propagation on the sensing capability.

Thank you so much for this note. The first part is used two antennas one for the sensor node and other one for the reference node. The second part is used the reference and the sensor node in same antenna. Both antennas are measured in the same place. Therefore, the refracted signals are considered in both parts because the refracted signals are affected in the sensor node and reference node. Moreover, the resistive sensor is used for short range applications.

3.)     The text needs improvements: a) instead of word "Ohms" use Greek letter, b) variables are to be in italic, c) p. 4, r. 152 - "MHz" is missing behind the number 937.

Thank you for these helpful comments. In the revised paper,

we have corrected these errors.

4.)    Further, the term "self-power" in the text seems to me unjustified and redundant independently on the fact that it has already been used by the authors in ref. [12-13]. The antenna tag does not power itself. It is powered by the energy of incident electromagnetic wave. Established RFID terminology uses for such transponders the term "passive" transponders; see e.g. ref. [1-3], [6-7]. Please justify the use of the proposed term or avoid it.

Thanks you for these important notes. We have deleted this error “self-power” from the revised paper.

Reviewer 2 Report

The authors have proposed similar ideas (RFID tag for sensor applications) and this cannot be considered a novelty of this work. Moreover, the term "miniaturized" is misleading. Perhaps tha patch antenna has been shrinked in size but other antennas for UHF RFID tag are smaller.

Author Response

Responses to the reviewers’ comments for the manuscript
Miniaturized Multiport Microstrip Patch Antenna Using Metamaterial for Passive UHF RFID-Tag Sensor Applications

We would like to thank the reviewers for their insightful comments regarding our manuscript. Their valuable input has helped us to improve its quality and readability. We have addressed all the reviewers’ comments in this revised version; our detailed responses to the reviewers’ comments are presented below.

REVIEWERS’ COMMENTS:

Reviewer 2:

The authors have proposed similar ideas (RFID tag for sensor applications) and this cannot be considered a novelty of this work. Moreover, the term "miniaturized" is misleading. Perhaps tha patch antenna has been shrinked in size but other antennas for UHF RFID tag are smaller.

We have compared our work with previous works, in terms of miniaturization ratio that have used the same technique [1]-[5]. The substrates are constructed from an artificial metamaterial unit cell reported in [1]- [5]. The magneto-dielectrics, presented in [1]- [3] are bulky and cannot be embedded with planar circuits. The substrates are presented in [4]- [5] thin enough to use in planar structure but the miniaturization ratio about 60 %. In our work, the proposed antenna can be easily used in a planar structure. Moreover, the miniaturization ratio is the highest ratio compared with [1]-[5], it is about 75 %.

[1]            H. Mosallaei, and K. Sarabandi, “Design and Modeling of Patch Printed on Magneto-Dielectric Embedded-Circuit Metasubstrate” IEEE Trans.  Antennas propag., vol. 55, no 1, pp. 45-52, Jun.2007.

[2]            P. Mookiah, and K. Dandekar, “Metamaterial-Substrate Antenna Array for MIMO Communication System” IEEE Trans.  Antennas propag., vol. 57, no 10, pp. 3283-3292, Oct.2009.

[3]            A. Pinsakul, S. Chaimool, P. Akkaraekthalin, “Miniaturized microstrip patch antenna printed on magneto-dielectric Metasubstrate,” Electrical Engineering/Electronics, Computer, Telecommunications and Information Technology (ECTI-CON), 2012 9th International Conference on, Phetchaburi, pp. 1- 4, 2012.

[4]            F. Farzami, K. Forooraghi, and M. Norooziarab, “Miniaturization of a Microstrip Antenna Using a Compact and Thin Magneto-Dielectric Substrate” IEEE Antennas wireless Propag.  Lett., vol. 10, pp. 1540-1542, 2011.

[5]            S. Kumar, D.K. Vishwakarma, “Miniaturization of microstrip patch antenna using an artificial planar magneto-dielectric meta-substrate,” IET Microw. Antennas Propag, vol.10, no.11, pp. 1235–1241, 2016.

Round  2

Reviewer 1 Report

The responce to the main previous concern #1 is still not satisfying and an issue of proving metamaterial character of the substrate remains. The values of effective parameters (permitivity and permeability) of claimed "metamaterial" need to be extracted either from equivalent or EM simulation  model and dispersion diagram should be presented to evalute a propagation constant in proposed artificial structure. This part of the paper is still not technically sound enough and I can not recommend the paper for publication.

Author Response

Miniaturized Multiport Microstrip Patch Antenna Using Metamaterial for Passive UHF RFID-Tag Sensor Applications

Reviewer 2 Report

I still think that the work is not very original (compared to same authors' paper) but that it could be of interest for some readers.

The reduction of antenna size comes to the cost of additional dielectric layers that will rise the cost of the final tag. This should be mentioned in a brief comment.

Author Response

Miniaturized Multiport Microstrip Patch Antenna Using Metamaterial for Passive UHF RFID-Tag Sensor Applications

Round  3

Reviewer 1 Report

The issue of proving effective parameters has been improved a bit according to my previous comments. When authors use any method to extract effective parameters of artificial media they usually provide brief basics of the method used for convenience of readers. That is why I checked "can be improved" three times.

Further Greek letter "omega" is still missing at p. 4, r. 137.